



# Reflection of low-frequency fast magnetosonic waves at the local two-ion cutoff frequency: Observation in the plasmasphere

Geng Wang[1,2], Mingyu Wu[1], Guoqiang Wang[1], Sudong Xiao[1], Irina Zhelavskaya[3,4], Yuri Shprits[3,4], Yuanqiang Chen[2], Zhengyang Zou[5], Zhonglei Gao[6,2], Wen Yi[2], and Tielong Zhang[7]

[1]Harbin Institute of Technology, Shenzhen 518055, China
[2]Key Laboratory of Geospace Environment , University of Science and Technology of China, Chinese Academy of Sciences, Hefei 230026, China
[3]Helmholtz Centre Potsdam, GFZ German Research Centre for Geosciences, Potsdam, Germany
[4]Institute of Physics and Astronomy, University of Potsdam, Potsdam, Germany
[5]State Key Laboratory of Lunar and Planetary Sciences, Macau University of Science and Technology, Macao 999078, PR China
[6]School of Physics and Electronic Sciences, Changsha University of Science and Technology, Changsha 410114, China
[7]Space Research Institute, Austrian Academy of Sciences, Graz 8042, Austria

**Correspondence:** Geng Wang (wanggeng@hit.edu.cn); Tielong Zhang (tielong.zhang@oeaw.ac.at)

**Abstract.** We investigate the reflection of low-harmonic fast magnetosonic (MS) waves at the local two-ion cutoff frequency ($f_{\mathrm{cutHe+}}$). By comparing the wave signals of the two Van Allen Probes, a distinct boundary where wave energies cannot penetrate inward are found in time-frequency domain. The boundary is identified as the time series of local $f_{\mathrm{cutHe+}}$. For a certain frequency, there exists a spatial interface formed by $f_{\mathrm{cutHe+}}$, where the incident waves should be reflected. The waves

with small incident angles are more likely to penetrate the thin layer where the group velocity reduces significantly, and being trapped in a period of several to tens of seconds before the reflection process complete. The cutoff reflection scenario can explain the intense outward waves observed by Probe-A. These results of MS reflection at $f_{\mathrm{cutHe+}}$ may help to predict the global distribution of MS waves and promote the understanding of wave-particle dynamics in the radiation belt.

## 1   Introduction

Fast magnetosonic (MS) waves are a type of electromagnetic emission commonly observed in the radiation belts. These waves generally range from the hydrogen gyrofrequency ($f_{\mathrm{H+}}$) to the lower hybrid resonance frequency ($f_{\mathrm{LH}}$) (Boardsen et al., 1992) and often exhibit harmonic structures at approximately the integral times of $f_{\mathrm{H+}}$ in high-resolution spectra (Gurnett, 1975; Perraut et al., 1982; Balikhin et al., 2015). It is suggested that the free energy of the ring current hot protons excites the harmonic waves via the ion Bernstein instability (Gary et al., 2010; Liu et al., 2011; Min et al., 2018; Liu et al., 2018a). In the

inner magnetosphere, MS waves normally have quasi-linear polarization (Curtis and Wu, 1979; Němec et al., 2005; Laakso et al., 1990) and have quasi-perpendicular wave vectors that account for the equatorial confinement (Russell et al., 1970; Zou et al., 2019). In recent years, MS waves have been considered to have the potential for heating electrons between the keV and MeV energy ranges via the Landau resonance (Horne et al., 2007), the transit time scattering (Summers and Ma, 2000; Bortnik





and Thorne, 2010) or the bounce resonance (Shprits, 2016; Tao and Li, 2016). Similar mechanisms can also result in a parallel

acceleration and can lead to the butterfly pitch angle distribution for MeV electrons (Xiao et al., 2015; Li et al., 2016b, a; Lei et al., 2017; Yang et al., 2017).

Recent studies found that the frequency of the wave peak occurrence increases from approximately 2 $f_{H+}$ at 2 $R_E$ to 21 $f_{H+}$ at 5 $R_E$ (Boardsen et al., 2016). Previous observations have demonstrated the propagation of MS waves in the radiation belt (Santolík et al., 2002; Su et al., 2017). The high harmonics with a few hundred Hz that are generated in the plasmasphere, can

penetrate into the low altitude of approximately 700 km (Gulelmi et al., 1975; Santolík et al., 2016). Additionally, Chen and Thorne (2012) considered the perpendicular propagation in the central symmetrical medium and predicted the trapping of MS waves in the vicinity of the plasmapause, which has been supported by subsequent observations (Ma et al., 2014; Yuan et al., 2019). A systematic study by Posch et al. (2015) revealed a wide MLT distribution, both inside and outside the plasmapause. Furthermore, Liu et al. (2018b) found that the multiple fine-scale of density irregularities where the WKB approximation is

violated also effectively blocked MS propagation. Teng et al. (2019) further revealed a wide distribution of MS waves below $f_{H+}$, with a higher intensity in the high-density afternoon section.

Theoretically, if the background plasma consists of at least two ion components, then the R-X mode waves below $f_{H+}$ should approach the cutoff frequency, which is dependent on the ion abundance ratios and magnetic field, and is independent on either wave normal angle or plasma density (Smith and Brice, 1964; Stix, 1992). The refractive index and wavenumber

are close to zero near the cutoff frequency, and thus the wave should be reflected as a consequence of Snell' law, as seen by the low-frequency hiss observed in the high latitude region (Gurnett and Burns, 1968; Santolík and Parrot, 1999; Chen et al., 2017). To date, there is a lack of evidence for the reflection of MS waves near the local two-ion cutoff frequency (or the helium cutoff frequency, referred to as $f_{cutHe+}$) in the outer radiation belt (or in the vicinity). We present direct evidence for such a reflection process in the current study.

## 2    Data and Methods

The data of two Van Allen Probes were used to analyze the wave behavior (Mauk et al., 2013). 64 Hz magnetic field data from the triaxial fluxgate magnetometer (MAG) of the Electric and Magnetic Field Instrument and Integrated Science (EMFISIS) suite (Kletzing et al., 2013) were processed through the fast Fourier transform (FFT, without detrending) and using the method of singular value decomposition (SVD) (Santolík et al., 2003), to obtain the wave normal angle and wave ellipticity. The 32 Hz

electric field data from the Electric Field and Waves (EFW) instrument (Wygant et al., 2013) were used. As there are only two available components, the third component was estimated based on $\mathbf{E} \cdot \mathbf{B} = 0$. The FFT was performed on the electric field and magnetic field (resampled to 32 Hz) to obtain the cross-power spectra, and the Poynting vector was then obtained (Santolík et al., 2010).

The electron density $n_e$ which was derived from the upper hybrid frequency (Kurth et al., 2015) measured by EMFISIS

HFR, the ambient magnetic field $B_0$ which was measured by EMFISIS MAG, and the proton ($H^+$) flux data measured by the Helium, Oxygen, Proton, and Electron (HOPE) instrument (Funsten et al., 2013) and by the Radiation Belt Storm Probes Ion



Composition Experiment (RBSPICE) instrument (Mitchell et al., 2013) were used for the calculation of growth rates.. The RBSPICE fluxes were divided by a factor of 3 to eliminate the mismatch with HOPE fluxes (Min et al., 2017; Kistler et al., 2016). Following the works of Kennel (1966) and Chen et al. (2010b), we can evaluate the wave temporal growth rate:

$$\gamma = \frac{D^{(1)}}{\partial D^{(0)}/\partial \omega},\tag{1}$$

and the corresponding convective growth rate:

$$K = \frac{\gamma}{|\mathbf{V_g}|}.\tag{2}$$

Here, $D^{(0)}$ and $D^{(1)}$ are the real and imaginary parts of the dispersion relation (Chen et al., 2010b, a). $D_s^{(1)}$ depends on the phase space density $F$ and its deviation with respect to energy and pitch angle. The technique details follow Su et al. (2018); Liu et al. (2018a); Wang et al. (2019).

Simulations were carried out based on a raytracing code, where the transformation of the coordinate systems followed Horne (1989), and the adaptive step size was derived by restricting the deviation between the locally calculated wavenumber and the integrated wavenumber. Two kinds of plasma conditions were used for comparison. The first condition combined the Plasma density in the Inner magnetosphere Neural network-based Empirical (PINE) density model (Zhelavskaya et al., 2016, 2017), and the TS05 magnetic field model (Tsyganenko and Sitnov, 2005). In the second condition, the observed $n_e$ and $B_0$ were simply replicated in the MLT to obtain the central symmetric two-dimensional distributions.

## 3 Event on 01 March 2017

### 3.1 Observation of the Wave Reflection

Figure 1 gives an overview of the MS wave event on 01 March 2017. At 11:35 UT, both Probe-A and Probe-B started to receive two groups of intense hydrogen band wave signals with several continuous patches: one approximately $4.0 - 7.0$ Hz and the other approximately $8.0 - 14.0$ Hz. The doubles of the peak frequencies of each wave patch in the lower frequency group match well with the peak frequencies of wave patches in the higher frequency group (black and white diamonds in 1a and 1e), indicating that the two groups of waves belong to the first harmonics and second harmonics, respectively. The frequency of each patch increases slightly within half an hour, the time scale of which is significantly larger than the rising tone structure (Fu et al., 2014), indicating its association with the injection of the wave sources, rather than nonlinear generation. A sharp low boundary can be found just above the helium gyrofrequency ($f_{He+}$), indicating the existence of the $f_{cutHe+}$. From Figure 1b and 1f, the plasma density $n_e$ of approximately 2000 cm$^{-3}$ indicates the immersion of probes inside the deep plasmasphere. From Figure 1c and 1g, the enhancements of hot H$^+$ (tens of keV) starting at 12:00 UT (beyond $L \sim 3.5$). These hot H$^+$ can lead to a positive convective growth rate for the low harmonics, in particular for the first harmonics in the range $4.0 - 7.0$ Hz (Figure 1d), at a quasi-perpendicular wave normal angle ($\psi = 89.5°$ here). This indicates a potential source region for the

**Figure 1.** Overview of the MS wave event on 01 March 2017. (a,e) Magnetic field power spectral density PSD. White dashed lines trace the local gyrofrequencies of proton ($f_{cH+}$), helium ($f_{cHe+}$) and oxygen ($f_{cO+}$). The black diamonds mark the frequencies of the peak PSD for each patch of the fundamental waves, while the frequencies of the white diamonds are double those of the black diamonds. (b,f) Electron number density $n_e$. (c,g) Energy-dependent $H^+$ differential fluxes $j_{H+}$ at a $90°$ pitch angle. (d) Convective growth rates for MS waves at a $89.5°$ wave normal angle. (h) Global distribution of the electron number density from PINE model, in the equatorial plane of the SM coordinate. The satellite trajectories are drawn as thick dashed and solid lines.



**Figure 2.** Time-frequency distributions of wave properties. (a,h) Magnetic field power spectral density PSD. The red dotted lines mark the lowest boundary (potential helium cutoff frequencies $f_{\mathrm{cutHe+}}$) of the visible waves. The dashed lines trace the local $H^+$ and $He^+$ gyrofrequencies. The grey closed curves mark the three selected regions in time-frequency domain. (b,i) Wave normal angle $\psi$ (unifying the two field-aligned orientations). (c,j) Magnetic ellipticity $\epsilon_B$. (d,k) Azimuthal angle of the Poynting flux $\phi_S$. $0°$ represents away from the Earth. The angle increases in a counterclockwise direction. (e,l) Radial orientation of the Poynting flux. (f,m) Azimuthal orientation of the Poynting flux. (g) Refractive index RI for the MS mode (at $\psi = 90°$). (n) Group velocity $V_g$ for the MS mode (at $\psi = 90°$).







**Figure 3.** Quantitative comparison of wave properties between Probe-A and Probe-B. (a,c) Total magnetic field power $P$. (b,d) Spectrum-weighted averaged azimuthal angle of Poynting flux $\overline{\phi}_S$.





waves observed in the inner $L$-shell. No waves can be observed in the region of positive convective growth rates, as the values of growth rates are in the small magnitude of approximately $10^{-7}$. This region may be close to the source region of the waves, and the free energy have been significantly released at the time the satellite arrived. During the wave period, both probes were located at $|\text{MLAT}| \sim 0.5° - 8.0°$, MLT$\sim 15.0 - 17.0$ and $L \sim 2.2 - 4.0$, i.e., near the equatorial plane of the afternoon sector.

Figure 1h presents the global distribution of the density predicted by PINE model, and the satellite trajectories during 11:30 UT $- 12:40$ UT, in the equatorial plane of the solar magnetosphere (SM) coordinate. The separation of the two satellites is less than $0.5R_{\text{E}}$ during the wave period, allowing a study of wave propagation in a small scale by comparative observation.

  The main parameters of properties for the first harmonics are presented in Figure 2. In general, the patterns of the intense waves in time-frequency domain are similar for both probes (Figure 2a and 2h), indicating the source region of the waves

recorded by the two probes were approximately the same (otherwise the patterns should be significantly different). The wave normal angles $\psi$ are larger than $60°$ (Figure 2n and 2i), and the ellipticities $|\epsilon_B|$ are less than 0.3 (Figure 2c and 2j), further confirm that they are MS waves. From Figure 2d and 2k, distinct differences in the azimuthal angles of the Poynting flux $\phi_S$ can be found for waves with different frequencies, suggesting different propagation paths for these waves. To investigate the cutoff reflection, three regions are selected in the time-frequency domain.

As shown by the red dotted lines in Figure 2a and 2h, there exist an obvious boundary at the lowest frequencies of the recognizable waves (PSD threshold is $10^{-2}$ nT$^2$Hz$^{-1}$) for each probe, with a constant proportion of the local $f_{\text{cH+}}$ ($\sim 0.26$ times). Region-I is determined by the two boundaries observed by the two probes. Within Region-I the moderate waves were detected by Probe-B but no signals were detected by Probe-A. This phenomenon was most probably a consequence of the reflection near the cutoff frequencies, i.e., near the observed boundaries. Assuming a two-ion plasma, the ion abundance

ratios can therefore be estimated ($\eta_{\text{H+}} = 98.5\%$ and $\eta_{\text{He+}} = 1.5\%$ here), and the dispersion relations can be obtained (for reasonability of this approach please see Appendix A. The distributions of refractive index RI=c$k/\omega$ and wave group velocity $V_g = d\omega/dk$ are then calculated. As shown in Figure 2g, RI decreases sharply near the predicted cutoff frequencies, which should lead to the significant deflection of the wave vector according to Snell's law. From 2n, there exists a narrow layer where $V_{\text{g}}$ decreases, which will substantially restrict the wave propagation.

Region-II and Region-III are selected, according to the obvious difference in the direction of the Poynting flux for each probe (Figure 2d and 2k). For Region-II, most of the waves of Probe-A were oriented westward (Figure 2f), while the waves of Probe-B were mainly oriented eastward (Figure 2m). This suggests that the upstream waves were between the MLT of each probe. For Region-III, in particular before 12:00 UT, the prominent outward and westward waves were measured by Probe-A (Figure 2e and 2f), while the inward and westward waves dominate the signals of Probe-B (Figure 2l and 2m). This suggests

that the upstream waves were in east of both probes, and a substantial amount of waves had experienced reflection before they were captured as outward signals by Probe-A.

  The behaviors of the waves in Region-II and Region-III are analyzed quantitatively in Figure 3. In Region-II, the intense inward waves of Probe-A have the values of $\overline{\phi}_S$ (the spectrum-weighted averaged $\phi_S$) are approximately $120° - 180°$, while the inward and outward waves of Probe-A have the values of $\overline{\phi}_S$ approximately -30°$-$ -130°. During most periods in the





beginning of Region-III, Probe-B exhibits domination of the inward waves ($\overline{\phi}_S \sim 120° - 180°$) in total wave powers, while the opposite results are exhibited in Probe-A ($\overline{\phi}_S \sim 0° - 60°$ for intense outward waves).

## 3.2 Simulations

For a further demonstration of the wave reflection process, raytracing simulations are performed, with three assumptions for simplification: First, the rays are confined within the equatorial plane, as the waves are observed to be no higher than

$|\text{MLAT}| \sim 8°$ with $\psi > 60°$. Second, two-ion plasma ($H^+$ and $He^+$) is considered when calculating the dispersion relation (98.5% $H^+$ and 1.5% $He^+$ here, as discussed in Section 3.1). Third, the direction of the wave vector and group velocity are treated as the same (proof in Appendix B).

The left column provides the simulations of the propagation scenario for the waves in Region-I. Multiple rays with different initial azimuthal angles $\phi$ are launched from a potential source point. The model magnetic field and density are used in the

simulation in Figure 4a, while the results using observed field and density (of Probe-B) are presented in Figure 4b. In both simulations, the rays with small incident angles (approximately equivalent to the angle between ray direction and radial axis which is normal to the contour of RI) experience the sharp reflection near the cutoff frequency where the value of RI decreases to zero, while the rays with intermediate incident angles experience relatively moderate reflection with the decreasing RI which is caused by the increasing magnetic field $B_0$. The difference between the two simulations appears for the rays with

large incident angles. Two rays with obvious separation in Figure 4b are selected and analyzed in Figure 4c−4h. As there exist a density dip structure at $R \sim 4.0$, $k$ decreases rapidly and lead to a rapid deflection of $\phi$ for both rays, the ray with a relatively larger incident angle (thick solid line) is therefore reflected, while the ray with a relatively smaller incident angle (thick dashed line) does not approach $\phi = 270°$ before escaping from the density dip, and continue to propagate inward.

The middle column provides the simulations for Region-II waves, using the observed $B_0$ and $n_e$. According to previous

results, rays are reverse traced from Probe-A/Probe-B to the east/west to find the source, and are forward traced from Probe-A/Probe-B to the west/east to reveal the reflection, as shown in Figure 4i/4j. Similar to the results in Region-I, the rays with smaller incident angles are reflected sharply near the cutoff frequency, while the rays with intermediate or large incident angles are reflected moderately in the outer position. A ray with the critical incident angle is selected (thick green line) and analyzed in Figure 4k−4p. At the beginning (group time $< 15$ seconds), as the increase of $B_0$, $k$ decreases gradually, leading to the

moderate deflection of $\phi$. The substantial reflection takes place at group time $\sim 20$ seconds, along with the slight reduction of $V_g$, indicating that the ray propagates into the narrow layer where $V_g$ decreases (Figure 2h). From the dispersion relation (Figure A1), $V_g$ and $k$ dramatically reduce with $f/f_{\text{cH+}}$ near $f_{\text{cutHe+}}$, which is a typical effect near the cutoff frequency.

The reflection scenario in the simulation of Region-III (right column) is similar to that of Region-II. Comparing Figure 4q and 4r, for the rays that can approach the satellite, the MLT extension of the source for the rays being reflected (Figure 4q) is

larger than that without reflection (Figure 4r). This can explain why some outward waves measured by Probe-A are dominant in Region-III. For the further revelation of the cutoff reflection process, a ray with a small incident angle (thick blue line) is analyzed in Figure 4s−4x. At the beginning (at group time $< 18$ seconds), neither the variation of $n_e$ nor that of $B_0$ deflect $\phi$ obviously. $V_g$ increases with $B_0$ as a consequence of the increasing of the slope of the dispersion curve (Figure A1). At group



**Figure 4.** Raytracing simulations in the $\mathbf{X} - \mathbf{Y}$ (equatorial) plane of the SM coordinates. The left, middle and right columns show respectively the simulations at three different points in time-frequency domain. The first two rows show the paths of the rays that: (a,b) launched at a fixed source, (i,q) pass through Probe A and (j,r) pass through Probe B. The rainbow colorbars in each panel represent the azimuthal angles of the rays at the corresponding intersection point of rays. The black-white colorbars represent the refractive indexes of the background plasmas. The arrows denote the directions of the rays. Model magnetic field and density are used in (a), while the observed magnetic field and density are used in (b,i−j,q−r). The positions of two probes are marked by triangles. In the second row, the thicker lines denote the selected rays which undergo analyses in the last six rows: the time series of (c,k,s) radial distance $R$, (d,l,t) wavenumber $k$, (e,m,u) group velocity $V_g$, (f,n,v) local azimuthal angle $\phi$, (g,o,w) electron number density $n_e$ and (h,p,x) ambient magnetic field $B_0$. The blue shaded areas in the middle and right columns represent the region of velocity drops.





time $\sim 18$ seconds, the ray starts to be trapped in the region where $V_\mathrm{g}$ decreases (shaded region in Figure $4s - 4x$). Only the
value of $k$ maintains the rate of decreasing with time, leading to the reflection of $\phi$, and the ray finally escape from the region
where $V_\mathrm{g}$ decreases after being trapped approximately 20 seconds.

## 4   Event on 20 July 2015

The second event was observed on 20 July 2015. From Figure 5, the waves display coherent harmonic structures, indicating a
relatively narrow source region. The first harmonic recorded by Probe-B is below local $f_{\mathrm{H+}}$ at $L \sim 5.0-5.6$. Meanwhile, Probe-
A was more inward than Probe-B, and thus observed the first three harmonics below $f_{\mathrm{H+}}$. Similarly with the previous event,
the vanishing of the first harmonic just above the local $f_{\mathrm{cutHe+}}$ is noticeable, comparing the spectra of Probe-A with Probe-B
at 14:30 UT, suggesting the potential reflection of waves. Most of the waves of Probe-A demonstrate a westward orientation,
while the waves of Probe-B demonstrate a eastward orientation. The outward waves gradually overwhelm inward waves at
larger MLTs (eastward) at locations of Probe-B. These phenomena are further demonstrated in the quantitative analysis in the
last two panels. Most waves observed by Probe-A have $\overline{\phi}_S$ below zero, while most waves of Probe-B have $\overline{\phi}_S$ above zero.
Considering that Probe-A was at an MTL $\sim 15.0$ and Probe-B was at an MTL $\sim 18.0$, the source region may be located at an
MTL between 15.0 and 18.0 in an outer $L$-shell. As Probe-B moved eastward and the source drifted westward, the observed
waves are gradually dominated by outward Poynting fluxes, which are most likely to be reflected in the vicinity of the local
$f_{\mathrm{cutHe+}}$ at $L \sim 3.9$.

## 5   Conclusions and Discussions

Low-harmonic MS waves are frequently observed near the radiation belts (Balikhin et al., 2015; Boardsen et al., 2016; Teng
et al., 2019) and have a potentially important dynamic influence on relativistic electrons (Maldonado et al., 2016; Yu et al.,
2019). The propagation of low-harmonic MS waves is thus important because it controls the wave distribution. In the present
study, the reflection of low-harmonic MS waves in the vicinity of the local two-ion cutoff frequency $f_{\mathrm{cutHe+}}$ within the plas-
masphere is studied. The results can be summarized as follows:

1. In the event on 01 March 2017, several wave patches were identified as the first harmonic MS waves. In a time-frequency
region (Region-I) which was below the local $f_{\mathrm{cutHe+}}$ for Probe-A and above local $f_{\mathrm{cutHe+}}$ for Probe-B, the waves
were intense for Probe-B but vanished for Probe-A. According to calculation and simulation, for waves with a certain
frequency, there exists an interface in space at $f_{\mathrm{cutHe+}}$, where the refractive index (RI) decreases to zero, and incident
175       waves should be reflected as a consequence of Snell's law. There also exists a spatial layer within which the group
velocity decreases significantly, and the waves with small incident angles should be trapped for several to tens of seconds
before the reflection process complete.



**Figure 5.** Wave properties and quantitative analysis of the Poynting flux for the event on 20 July 2015. (a,g) Magnetic field power spectral density PSD. (d,j) Azimuthal angle of the Poynting flux $\phi_S$. $0°$ represents away from the Earth. The angle increases in a counterclockwise direction. (e,k) Radial orientation of the Poynting flux. (f,l) Azimuthal orientation of the Poynting flux. (e,k) Total magnetic field power $P$. (f,l) Spectrum-weighted averaged azimuthal angle of Poynting flux $\overline{\phi}_S$.





2. In another time-frequency region (Region-III), prominent outward waves were measured by Probe-A, while the inward waves dominated the signals of Probe-B. Simulations suggested that a certain amount of the outward waves recorded by Probe-A were previously reflected near their cutoff frequencies.

3. The results also demonstrate that there exist three factors which will lead to the decrease of RI and thus the reflection of MS waves in the plasmasphere: The structure of density dip can separate the wave paths for the waves with smaller and larger incident angles. The increase of ambient magnetic field can lead to the gradual decrease of RI and thus reflect the waves with intermediate incident angles. The waves with small incident angles can penetrate deep into the plasmasphere and be reflected near the interface of the cutoff frequency.

4. In the event on 20 July 2015, some intense first-harmonic waves were observed by Probe-B above the $f_{\mathrm{cutHe+}}$ but were not measured by Probe-A below the $f_{\mathrm{cutHe+}}$ within the same time-frequency region. Outward waves were observed by both probes, and the out ward waves gradually overwhelmed the inward waves at larger MLT for Probe-B. These results strongly suggest wave reflection near the $f_{\mathrm{cutHe+}}$ at $L \sim 3.9$.

In Region-II, the azimuthal angels observed by Probe-B are almost perpendicular to the radial axis. The source of the rays with these angles are to the west of Probe-A from simulations. However, the observed upstream waves were in east of Probe-A. This may be due to the separation of waves by the density dip along MLT in reality, similar to the mechanism in Figure 4b.

Liu et al. (2018b) found that the multiple fine-scale density irregularities can block MS propagation. The event in the present study, however, penetrate deep into the plasmasphere, which may be because the waves are initially generated inside the plasmasphere and are not influenced by the multiple fine-scale structures. In fact, most of the waves near $f_{\mathrm{cutHe+}}$ are actually reflected before their wavelengths (minimum $\lambda \sim 300$ km for the case of small incident angle according to Figure 4t) become comparable with the scales of density irregularities (generally $> 0.05$ $R_{\mathrm{E}}$ according to Figure 1b and 1f).

For the inward propagated high-harmonic waves, with the increase in the ambient magnetic field or decrease in the plasma density, the refractive index decreases accordingly, and waves should be reflected, as studied by Chen and Thorne (2012); Ma et al. (2014); Yuan et al. (2019). For the low harmonics, the waves propagating inward in a direction close to the radial axis are more likely to penetrate deep in the plasmasphere, and be reflected by the absolute boundary formed by the local $f_{\mathrm{cutHe+}}$. As the cutoff reflection of the low-harmonic MS waves should be common, these results may help to predict the global distribution of MS waves and therefore promote the understanding of wave-particle dynamics in the outer radiation belt.

**Appendix A**

Here we demontrate the reasonability of neglecting the minor content of oxygen ion in deriving the MS mode dispersion relation. In the cold plasma, the following Stix parameters are helpful to investigate the dispersion relation (Stix, 1992):

$$L = 1 - \sum_s \frac{\omega_{\mathrm{p}s}^2}{\omega(\omega - \Omega_s)}, \ R = 1 - \sum_s \frac{\omega_{\mathrm{p}s}^2}{\omega(\omega + \Omega_s)}, \ P = 1 - \sum_s \left(\frac{\omega_{\mathrm{p}s}}{\omega}\right)^2, \tag{A1}$$



**Figure A1.** Dispersion relations around ion gyrofrequencies for perpendicular wave mode, under the conditions of $B_0 = 1180$ nT and $n_e = 1500$ cm$^{-3}$. (a) The plasma contains no O$^+$ ions. (b) The plasma contains 5% O$^+$ ions. For each case, the ratio of He$^+$ cutoff frequency to H$^+$ gyrofrequency $f_{\mathrm{cutHe+}}/f_{\mathrm{cH+}}$ is set to 0.261 (based on the event), and the abundance ratios of H$^+$ and He$^+$ are obtained accordingly. One can find that the difference in the MS mode dispersion curve (red curve) is small for the two sets of ion abundance ratios.





here $\omega_{\mathrm{p}s} = (n_s q_s^2/m_s \varepsilon_0)^{\frac{1}{2}}$ and $\Omega_s = q_s B/m_s$ are respectively the plasma frequency and the gyrofrequency of a particle species s.

The cutoff frequency $\Omega_{\mathrm{cut}}$ is the frequency where the phase velocity equals to zero (Smith and Brice, 1964), and can be obtained by setting the Stix parameter L equal to zero:

$$1 - \sum_s \frac{\omega_{\mathrm{p}s}^2}{\omega_{\mathrm{cut}}(\omega_{\mathrm{cut}} - \Omega_s)} = 0. \tag{A2}$$

Considering the charge neutrality condition, the determination of full ion abundance ratios requires the values of at least two characteristic frequencies (except gyrofrequencies) to be known in H$^+$, He$^+$ and O$^+$ plasma. However, if the O$^+$ abundance

$\eta_{\mathrm{O+}}$ is much lower than the H$^+$ abundance $\eta_{\mathrm{H+}}$ and if the focused mode belongs to the H$^+$ band which has a much larger frequency than the oxygen gyrofrequency $\Omega_{\mathrm{O+}}$, then we have the following relations:

$$-\frac{\omega_{\mathrm{pH+}}^2}{\omega(\omega - \Omega_{\mathrm{H+}})} \gg \frac{\omega_{\mathrm{pO+}}^2}{\omega(\omega - \Omega_{\mathrm{O+}})}, \ when \ \Omega_{\mathrm{He+}} < \omega < \Omega_{\mathrm{H+}} \tag{A3}$$

$$\frac{\omega_{\mathrm{pHe+}}^2}{\omega(\omega - \Omega_{\mathrm{He+}})} \gg \frac{\omega_{\mathrm{pO+}}^2}{\omega(\omega - \Omega_{\mathrm{O+}})}, \ when \ \omega \to \Omega_{\mathrm{He+}} \tag{A4}$$


$$\frac{\omega_{\mathrm{pH+}}^2}{\omega(\omega + \Omega_{\mathrm{H+}})} \gg \frac{\omega_{\mathrm{pO+}}^2}{\omega(\omega + \Omega_{\mathrm{O+}})}, \ when \ \Omega_{\mathrm{He+}} < \omega < \Omega_{\mathrm{H+}} \tag{A5}$$

$$\frac{\omega_{\mathrm{pH+}}^2}{\omega^2} \gg \frac{\omega_{\mathrm{pO+}}^2}{\omega^2}, \ when \ \Omega_{\mathrm{He+}} < \omega < \Omega_{\mathrm{H+}}. \tag{A6}$$

Therefore, if the three-ion (H$^+$, He$^+$ and O$^+$) plasma are approximated as two-ion (H$^+$ and He$^+$) plasma, i.e., the terms with

oxygen plasma frequency $\omega_{\mathrm{pO+}}$ are dropped in Equation (A1), the Stix parameters will have only a negligible change. Under such an approximation, a group of ion abundance ratios ($\eta_{\mathrm{H+}}$ and $\eta_{\mathrm{He+}}$) can be obtained by substituting the observed value of $\omega_{\mathrm{cutHe+}}/\Omega_{\mathrm{H+}}$ into Equation (A2). Consequently, the approximated dispersion relations for the modes in H$^+$ band can be found.

**Appendix B**

Here we provide the proof that the directions of the wave vector and Poynting flux are the same for a perpendicular MS wave. In the magnetized cold plasma, the frequency for a plane wave is the function of magnetic field $B_0$, electron density $n_e$, wave normal angle $\psi$ and wavenumber $k$:

$$\omega = \omega(B_0, n_e, \psi, k). \tag{B1}$$

Following Stix (1992), the group velocity can be expressed as:

$$\mathbf{v_g} = \frac{\partial \omega}{\partial \mathbf{k}} = \hat{\mathbf{k}}\frac{\partial \omega}{\partial k} + \hat{\boldsymbol{\psi}}\frac{1}{k}\frac{\partial \omega}{\partial \psi}, \tag{B2}$$





and the angular difference $\alpha$ of the two orthogonal directions can be expressed as:

$$\tan\alpha = \frac{\frac{1}{k}\frac{\partial\omega}{\partial\psi}}{\frac{\partial\omega}{\partial k}} = -\frac{1}{k}\frac{\partial k}{\partial\psi} = -\frac{1}{\mu}\frac{\partial\mu}{\partial\psi}. \tag{B3}$$

Here $\mu$ is the refractive index. Expressing the dispersion relation in the form:

$$\tan^2\psi = -\frac{P(\mu^2-R)(\mu^2-L)}{(S\mu^2-RL)(\mu^2-P)}, \tag{B4}$$

where $R, L$ and $P$ are the Stix parameters, and $S = (R+L)/2$. Considering a small angle $\beta$, and Expanding the left side of Equation (B4) around $\pi/2$, and considering that $\mu^2 \sim RL/S \ll P$ for the fast magnetosonic mode branch, we can obtain the following relation:

$$\tan\alpha = \frac{1}{\mu}\frac{\partial\mu}{\partial\beta} = -\beta\frac{\frac{RL}{S^2}-1}{2\beta^2\frac{RL}{S^2}-1}, \tag{B5}$$

here $\beta = \pi/2 - \psi$. As $\tan\alpha \sim 0$ when $\beta \sim 0$, the direction of group velocity and wave vector are the same for the perpendicular
MS waves.

*Author contributions.* GW conceived the study and wrote the manuscript. IZ and YS provided the estimated data of density distribution. ZG made contributions to wave analyses and ray-tracing code. All co-authors contributed to discussions and commented on the manuscript.

*Competing interests.* The authors declare that they have no conflict of interest.

*Acknowledgements.* The work in China was supported by NSFC grants 41774171, 41974205, 41774167, 41804157, 41904156 and 41904135.
The authors also acknowledge the financial support provided by Shenzhen Science and Technology Research Program JCYJ201803306171918617 and Shenzhen Science and Technology Program (Group No. KQTD201804410161218820), and by Key Laboratory of Geospace Environment, Chinese Academy of Sciences, University of Science and Technology of China. Tielong Zhang was supported by CAS Center for Excellence in Comparative Planetology. We appreciate EMFISIS, ECT, EFW and RBSPICE teams for the use of Van Allen Probes data. Data are available from the following websites: http://emfisis.physics.uiowa.edu/Flight/, http://www.RBSP-ect.lanl.gov/, http://www.space.umn.
edu/rbspefw-data/ and http://rbspice.ftecs.com/Data.html. The magnetic field model of TS05 can be obtained from http://geo.phys.spbu.ru/%7etsyganenko/modeling.html.



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
