# Peer review of "Reflection of low-frequency fast magnetosonic waves at the local two-ion cutoff frequency: Observation in the plasmasphere"

_Annales Geophysicae, 2021_

## Author Response (AR1)

Response for the comments of     Referee #1

We thank the referee for the comments and suggestions.

1. Comment: My only substantial concern with the work in the paper is that the satellites have significant L* separation during the event on the 20thJuly 2015, it is unclear how the authors determined that the observations are of the same magnetosonic waves. The analysis of this event is also fairly limited in comparison to the rest of the paper and potentially does not add to the paper.

Reply: I agree with this concern. In the second case,   two probes observed the waves with the same frequencies at the same time, which indicate the same L-shell of the wave source. It is possible that the different waves emitted from the adjacent source areas and were received by the two probes.

The most useful information here is the reflected signals recorded by Probe B, which are more consistent and obvious than the first case. However, as no   cooperative observations   here   for the same waves or the waves with similar paths,          we have removed this case from the manuscript in the revised version.

2. Comment:   In line 108, the authors say that the waves in Region III observed by Probe A are westward, however it is difficult to see that directly in Figure 2f as there appears to be similar amounts of blue and red. Is this determined by a power weighed average? If so, this could be included in the text.

Reply: Thanks for the comment. In this case, some waves in Region III observed by Probe A have both westward and outward orientations (11:50—11:52 UT, 11:53—11:58 UT). These waves have prominently outward wave power, as shown in Figure 3a. We have now revised the text accordingly.

3. Comment:   Figure 4 panels (e)-(h), Figure 4 panels (k)-(p) and Figure 4 panels (s)-(x) are very small when printed and hard to read. I would suggest having these are a separate figure.

Reply: Thanks for the suggestions. The previous Figure 4   has now been separated into two figures.

We thank the referee again for the comments.

Response for the comments of    Referee #2

We thank the referee    for the comments and suggestions.

Major comment 1:    In the simulation results shown in Figures 4s-4x, the ray slowed down near the reflection point. It is not "trapping".

Reply: Thanks for pointing out that the use of 'trapping' may not be appropriate. The sentences in line 149-151 has been revised as 'the ray slows down in the region where $V_g$ decreases (shaded region in Figure 4s − 4x). Only the value of k maintains the rate of decreasing with time, leading to the reflection of φ, and the ray finally escape from the region where $V_g$ decreases after approximately 20 seconds.'

Major comment 2:    The "Conclusion and Discussion" section is basically descriptions of the two event studies. I suggest the authors briefly summarize the two event studies and draw more generalized conclusions.

Reply: Thank you for the suggestion.    By taking into consideration of the comments of another referee, we are considering to remove the second case from the article, because the scenario of this case is generally the same with the Region-I and Region-II in the first case.    The forth conclusion is therefore removed, and the revised conclusions are less redundant.

Reply for minor comments: Thanks for these comments. The manuscript has been revised accordingly.

Minor issues:

  Line 35:   "as seen by the low-frequency hiss observed in the high latitude region" : this sentence does not make sense.

Reply:

This has been revised as "Previous studies have demonstrated    the reflection of hiss waves in the high latitude region."

Minor issues:

Line 89: for both probes - > at locations of the two probes.

Reply:

This has been revised.

Minor issues:

Line 101: a lack of the right bracket )

Reply:

This has been revised.

Minor issues:

Line 139: as the increase of B0 , k decreases gradually, leading to the moderate -> as

B0 increases, the k decreases gradually, leading to a moderate ⋯

Reply:

This has been revised.

Minor issues:

Line 153: The second event was observed on 20 July 2015 -> We show a second case

that occurred on 20 July 2015 to show the reflection xxx.

Line 153: display-> exhibit

Line 155: Similarly with - > similar to

Line 155-157: this sentence needs to be reorganized.

Line 157: Waves of Probe-A: Waves observed by Probe A

Reply:   Thanks   for the suggestions.   Because the scenario of the second event is generally the same with the Region-I and Region-II in the first event,   the corresponding paragraph has been removed along with the second event.

We thank the referee again for the comments.